# Autophagy differentially regulates TNF receptor Fn14 by distinct mammalian Atg8 proteins

Hila Winer[1], Milana Fraiberg[1], Adi Abada[1], Tali Dadosh[2], Bat-Chen Tamim-Yecheskel[1] & Zvulun Elazar[1]

Autophagy, a conserved membrane trafficking process, sequesters cytoplasmic components into autophagosomes and targets them for lysosomal degradation. The TNF receptor Fn14 participates in multiple intracellular signaling pathways and is strongly induced upon tissue injury and solid tumorigenesis. While Fn14 is a short-lived protein, the regulation of its levels is largely obscure. Here we uncover a role for autophagy in Fn14 turnover, wherein specific core autophagy Atg8 proteins play distinct roles: Fn14 accumulates in the ERGIC in absence of GABARAP but within endosomes in the vicinity of autophagic membranes in absence of GATE-16. Moreover, GABARAP regulates overall cellular levels of Fn14, whereas GATE-16 regulates TWEAK signaling by Fn14 and thereby NF-κB activity. These findings not only implicate different Atg8 proteins in distinct roles within the mechanism of selective autophagic regulation of Fn14, but may also provide a more general view of their role in mediating autophagosome biogenesis from different membrane sources.

[1] Department of Biomolecular Sciences, The Weizmann Institute of Science, Rehovot 76100, Israel. [2] Department of Chemical Research Support, The Weizmann Institute of Science, Rehovot 76100, Israel. These authors contributed equally: Hila Winer, Milana Fraiberg, Adi Abada. Correspondence and requests for materials should be addressed to Z.E. (email: zvulun.elazar@weizmann.ac.il)

Fibroblast growth factor inducible 14 (Fn14) is a tumor necrosis factor (TNF) receptor family member that is strongly upregulated in response to tissue injury[1,2] and is overexpressed in solid tumors[3–10]. It is activated by TNF-like weak inducer of apoptosis (TWEAK/TWK), a member of the TNF cytokines superfamily, that controls various cellular processes, including proliferation, migration, differentiation, apoptosis, and inflammation[1,11–13]. Activation of Fn14 by this ligand leads to recruitment of TNF receptor-associated factor 2 (TRAF2), an adaptor protein responsible for classical and alternative nuclear factor-κB (NF-κB) signaling pathways[1,2,11,12,14–20]. Fn14 is a short-lived protein (74 min half-life) that is constantly synthesized and its degradation is mediated by lysosomes in a ligand-dependent as well as independent manner[19]. The mechanism governing Fn14 turnover, however, remains poorly understood.

Autophagy is a process of bulk degradation of proteins and organelles that is induced under stressful conditions such as starvation, oxidative damage, and growth factor deprivation[21,22]. Selective cargo recruitment to the autophagosome is a pivotal function of autophagy essential to maintain cellular homeostasis[23,24]. Selective autophagy utilizes the core autophagy genes and varying sets of proteins for different cargos[25]. Mammalian Atg8 homologs constitute a family of proteins subdivided into two subfamilies: microtubule-associated protein 1 light chain 3 (hereafter referred to as LC3s) and γ-aminobutyric acid receptor-associated proteins (GABARAPs), are implicated in selective autophagy processes[26]. Autophagy is initiated by nucleation of an isolation membrane, which elongates into a cup-shaped phagophore that engulfs part of the cytoplasm and finally seals into a mature double-bilayer autophagosome that fuses with the lysosome[21,22,27]. Initial phagophore nucleation requires recruitment of various autophagic factors at different stages[21,22,28]. Among the first complexes recruited to the isolation membrane is the class III phosphatidylinositol 3-kinase (PI3K) complex that produces phosphatidylinositol 3-phosphate (PI3P), consisting of the catalytic subunit vacuolar protein sorting (VPS) 34, the myristoylated serine/threonine kinase p150, ATG14, and Beclin-1. PI3P consequently targets several proteins to the nascent autophagic membrane, including WD repeat phosphoinositide-interacting protein (WIPI) 1/2[21,22,28–31]. Autophagic activity also involves two ubiquitin-like conjugation systems—the ATG12~ATG5-ATG16L complex and conjugation of Atg8 proteins with phosphatidylethanolamine (Atg8-PE), the latter regulated by the cysteine protease ATG4[21,22,32]. In selective autophagy, cargo is recruited into phagophores by cargo receptors. For example, p62 (also termed Sequestosome-1) bridges ubiquitinated cargo through its Ub-binding domain with Atg8 proteins through its LC3-interacting region[21,22,24,28].

While autophagy is implicated in regulation of several receptors[33–36], direct implication of the autophagic machinery in regulation of TNF receptors has not yet been reported. Here we demonstrate that autophagy plays central roles in Fn14 function and turnover through differential regulation of this receptor by distinct activities of mammalian Atg8 proteins GABARAP and Golgi-associated ATP enhancer-16 (GATE-16).

## Results

**Autophagy regulates Fn14 turnover.** To identify the mechanism that governs the turnover of Fn14 we first inhibited both lysosomal and proteasomal degradation in HeLa cells by Bafilomycin A1 (BafA) and Velcade, respectively (Fig. 1a and Supplementary Fig. 1a). BafA treatment, with or without the Fn14 ligand TWK, led to accumulation of Fn14, whereas treatment with Velcade had no effect (Fig. 1a and Supplementary Fig. 1a), in consistence with

the notion that Fn14 is mainly degraded by the lysosome[12]. Accordingly, Fn14 also accumulated upon treatment with the well-known acidification inhibitors ammonium chloride (NH4Cl) or chloroquine (CQ)[37,38] (Supplementary Fig. 1b). Moreover, treatment with TWK induced further degradation of Fn14 that otherwise accumulated upon co-treatment with BafA, implying that lysosomal influx of Fn14 was higher in TWK-treated cells (Fig. 1a). We then followed the subcellular localization of Fn14 by confocal immunofluorescence microscopy. Inhibition of lysosomal degradation resulted in increased localization of Fn14 to perinuclear LAMP1-positive late endosomes and lysosomes (Fig. 1b and Supplementary Fig. 1c). To determine the localization of other, LAMP1-negative Fn14 subpopulations we examined several organelle markers (Fig. 1c, d and Supplementary Fig. 1c). Upon BafA treatment a discernible portion of Fn14 localized to the Golgi apparatus and endoplasmic reticulum (ER)-Golgi intermediate compartment (ERGIC) (labeled with GRASP65 and ERGIC-53, respectively), while no evidence of colocalization with the ER marker Calnexin was observed (Supplementary Fig. 1d). Fn14 was detected in neither immunoblot nor immunofluorescence analysis upon *Fn14* knockdown in HeLa cells, excluding nonspecific binding of the anti-Fn14 antibody (Supplementary Fig. 1e,f).

Delivery of membrane-bound proteins to the lysosome is typically mediated by either endosomal sorting complex required for transport (ESCRT)-mediated multivesicular bodies (MVBs) or autophagy[21,22,39]. To determine the role of MVBs in Fn14 degradation we transfected HeLa cells with siRNA against *TSG101*, a key ESCRT protein essential for MVB formation[40]. The knockdown of *TSG101* led to a TWK-independent decrease in endogenous Fn14 levels, partially rescued by BafA treatment, in sharp contrast to the accumulation of bona fide MVB substrate epidermal growth factor receptor (EGFR)[39,41] (Fig. 2a and Supplementary Fig. 2a,b). To address whether autophagy plays a role in Fn14 degradation key autophagic factors were knocked out or down. Knockout of *ATG5* in mouse embryonic fibroblast cells led, independently of BafA or TWK treatment, to an increase in ectopically expressed Fn14 (Fig. 2b). This treatment did not affect the level of the well-studied TNF receptor TNFR1 (Supplementary Fig. 2c), suggesting specificity toward Fn14. Furthermore, the combined knockdown of key autophagic factors *ATG7* and *ATG3* in HeLa cells and ovarian cancer cells (OVCAR8) led not only to TWK-independent inhibition of autophagic flux (assayed by LC3 lipidation) as expected, but also to accumulation of endogenous Fn14 (Fig. 2c and Supplementary Fig. 2d) without affecting EGFR levels (Supplementary Fig. 2e). Finally, accumulation of endogenous Fn14 was detected in clustered regularly interspaced short palindromic repeats (CRISPRs)/CRISPR-associated protein (Cas) 9 knockout cells of all Atg8 isoforms (Fig. 2g and Supplementary Fig. 2f), in which autophagy is inhibited[42].

To characterize the localization and cellular levels of Fn14 upon inhibition of autophagy we probed it by immunofluorescence microscopy (Fig. 2d, e) and immunoblotting (Supplementary Fig. 2g,h) while employing two different strategies to inactivate Atg8 proteins: knockdown of Atg8 isoforms and overexpression of the dominant-negative ATG4B (C74A) mutant (ATG4B^DN), which is known to prevent cleavage and lipidation of Atg8 proteins, thereby reducing the active pool of Atg8s[32,43]. Both strategies similarly yielded the appearance of large Fn14-positive puncta (Fig. 2d, e, white arrows) and a rise in Fn14 levels, located mainly in a perinuclear region and partially colocalizing with LAMP1 (Fig. 2d, e and Supplementary Fig. 2j). Similar phenotypes were observed upon depletion of the early acting autophagy-essential factor Beclin-1[28,30,31] (Fig. 2f and Supplementary Fig. 2i,k). On the other hand, induction of autophagy by

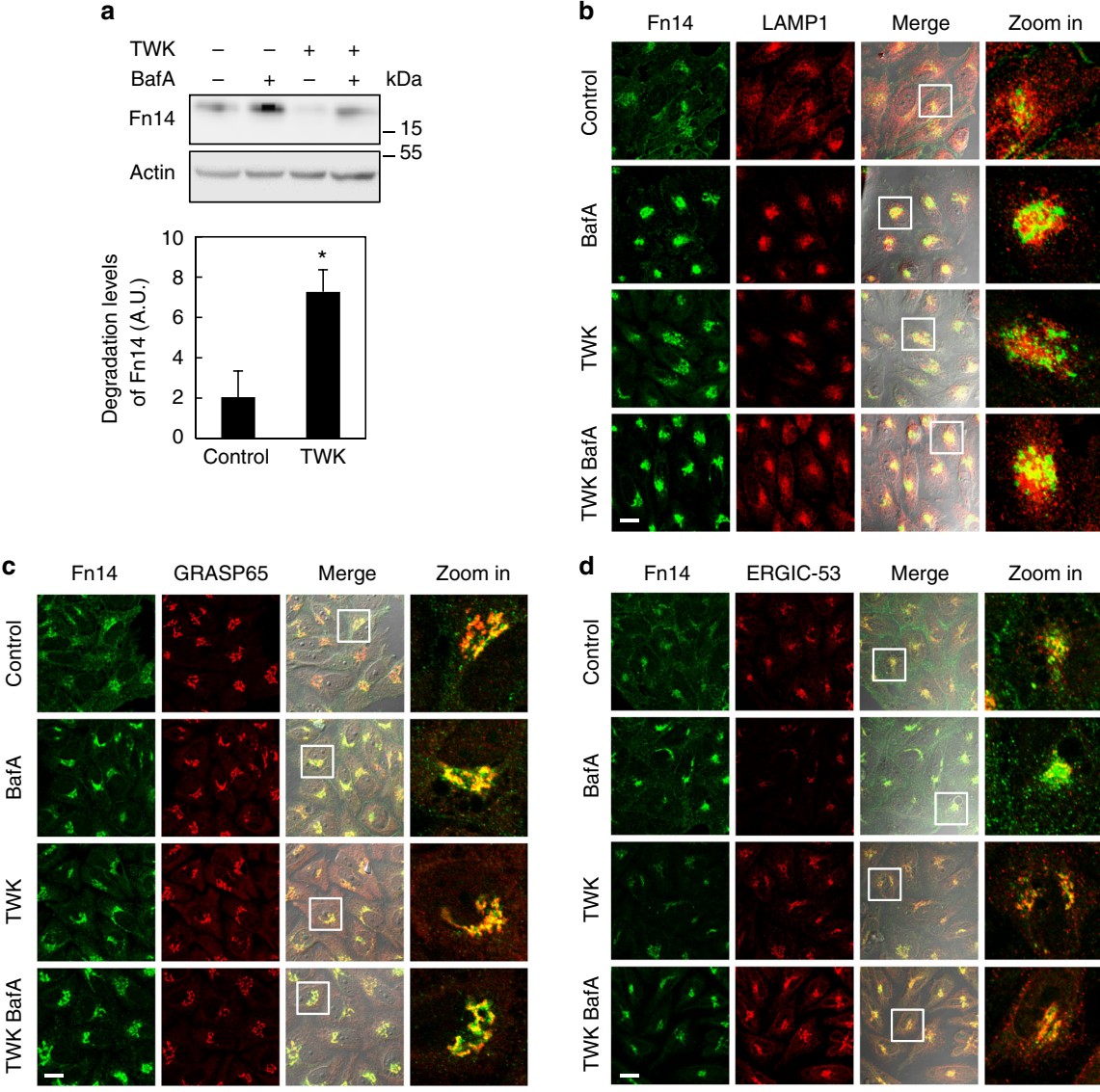

**Fig. 1** Fn14 is degraded in lysosomes and undergoes trafficking through Golgi. **a** HeLa cells were treated with BafA and TWK as indicated, extracted proteins were immunoblotted (upper panel) and lysosomal flux was calculated (lower panel) (*$P < 0.05$, $n = 7$ biological repeats). **b**–**d** Cells were treated as in **a**, immunostained for Fn14 and LAMP1 (**b**), GRASP65 (**c**), or ERGIC-53 (**d**), and analyzed by confocal microscopy (scale bar, 20 μm). Large magnification of stained cells is presented in the right column. Comparisons by $T$ test; mean ± s.e.m.

starvation (Earle's balanced salt solution (EBSS)) or mammalian target of rapamycin-inhibitor rapamycin led to a rise in lysosomal influx of Fn14 and a consequent reduction in its levels (Fig. 3a–d), while EGFR levels rather slightly increased (Fig. 3d). To determine whether Fn14 localizes to autophagosomes under basal conditions, HeLa cells stably expressing green fluorescent protein (GFP)-tagged Atg8s were subject to immunofluorescence analysis using either three-dimensional (3D) stochastic optical reconstruction microscopy (3D STORM)[44] or confocal microscopy. Indeed, Fn14 localized to autophagosomes labeled with GFP-tagged GABARAP, GATE-16, or LC3B (Fig. 3e and Supplementary Fig. 3). Taken together, our findings support the notion that autophagy sequesters Fn14 into autophagosomes for lysosomal degradation.

**Fn14-positive vesicles accumulate in a TWEAK-dependent manner**. To determine whether the accumulation of Fn14-positive vesicles upon inhibition of autophagy is ligand-

dependent we treated wild type (WT) or ATG4B[DN] cells with TWK and determined the localization of Fn14 and its adaptor protein TRAF2 by immunofluorescence. In WT cells Fn14 initially localized to the plasma membrane (Fig. 4a). Following 30 min of treatment with TWK, however, it was mostly internalized, whereas TRAF2 was mostly detected throughout the cytoplasm, and both were mostly degraded within 2 h (Fig. 4a). Inhibition of autophagy by ATG4B[DN] led to the accumulation of Fn14-positive vesicles that were also positive for TRAF2. Importantly, TWK treatment led to further accumulation of these vesicles in a time-dependent manner (Fig. 4a and Supplementary Fig. 4a) and to colocalization with early acting autophagic factors WIPI1 and p62[21,22,28] and early endosomal marker EEA1 (Supplementary Fig. 4b). Accordingly, biochemical analysis of ATG4B[DN] cells incubated with TWK demonstrated cofractionation of Fn14 with TRAF2, p62, and the endosomal proteins EEA1, Rab5, and Rab7 (Fig. 4b). These results imply that Fn14 accumulates in vesicles positive for both autophagic and endocytic factors. As Fn14-positive vesicles in ATG4B[DN] cells were also positive for TRAF2,

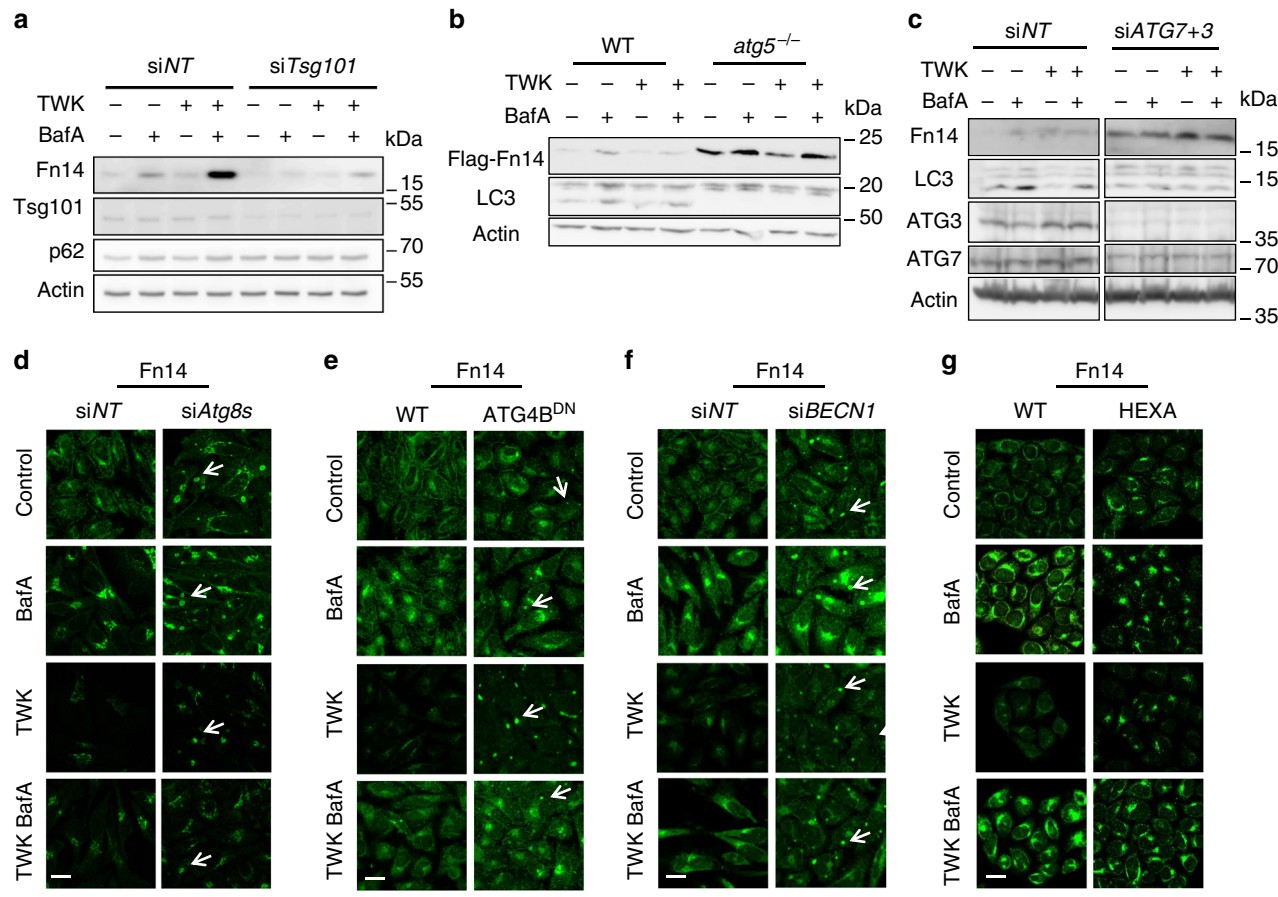

**Fig. 2** Inhibition of autophagy results in Fn14 accumulation and redistribution into large vesicular structures. **a–c** Transfected cells as follows were treated with BafA and TWK as indicated and extracted proteins were immunoblotted: **a** HeLa cells were transfected with non-targeting siRNA (si*NT*) or siRNA against *TSG101* (si*TSG101*); **b** WT and *ATG5*-knockout (*atg5*$^{-/-}$) MEFs were transfected with overexpression plasmid of Flag-Fn14; **c** HeLa cells transfected with si*NT* or siRNA against *ATG7* and *ATG3* (si*ATG7* + *3*). **d–g** HeLa cells as follows were treated with BafA and TWK as indicated, immunostained for Fn14, and analyzed by confocal microscopy (scale bar, 20 μm): **d** cells transfected with si*NT* or siRNA against all six Atg8 isoforms (si*Atg8s*); **e** WT or ATG4B$^{DN}$ stable expression cells; **f** cells transfected with si*NT* or siRNA against *BECN1* (si*BECN1*); **g** HeLa WT cells or CRISPR knockout of all Atg8s (HEXA)

we tested whether inhibition of autophagy affects Fn14 signaling to NF-κB. Substantial increase in NF-κB activity upon TWK treatment was observed in both WT and in ATG4B$^{DN}$ cells indicating that inhibition of autophagy sustains signaling of internalized Fn14 within vesicles (Fig. 4c). Supporting this notion, Fn14 was sensitive to proteinase K in WT and ATG4B$^{DN}$ cells (Fig. 4d). This indicates that upon inhibition of autophagy the receptor within accumulated puncta is exposed to the cytosol and therefore signaling. As a positive control to the assay, p62 was processed upon proteinase treatment in cells expressing ATG4B$^{DN}$, emphasizing the open state of its associated autophagic membranes (Fig. 4d).

We next examined the role of p62 in the autophagic degradation of Fn14. The knockdown of p62 led to accumulation of Fn14 in perinuclear regions and prevented its recruitment into ATG4B$^{DN}$-induced vesicles in a ligand-independent manner (Fig. 4f), accompanied by a general rise in cellular levels of Fn14 (Fig. 4e). These observations imply that accumulation of Fn14 in ATG4B$^{DN}$-induced vesicles is reversible and highly dependent on p62, and that efficient degradation of Fn14 requires p62.

**Autophagy and MVBs both contribute to Fn14 trafficking.** Our results so far indicate that Fn14 accumulates upon inhibition of autophagy, yet ligand-dependent lysosomal degradation of the receptor is still evident (Fig. 2 and Supplementary Fig. 2), a

process most likely promoted by the MVB pathway. Production of PI3P is essential both for the recruitment of autophagic factors to the site of autophagosome formation[22,28–31] and to endocytosis[45]. To examine whether PI3P is required for the accumulation of Fn14-positive vesicles, we treated WT and ATG4B$^{DN}$ cells with the PI3K inhibitors wortmannin and VPS34-IN2[46–48]. Inhibition of PI3K impaired the internalization of Fn14 in WT cells and its translocation into p62-positive vesicles in ATG4B$^{DN}$ cells (Fig. 5a and Supplementary Fig. 5a). To determine whether the accumulation of Fn14 in intracellular vesicles depends on TRAF2, we knocked down TRAF2 in WT and ATG4B$^{DN}$ cells. As depicted in Fig. 5b, knockdown of TRAF2 in ATG4B$^{DN}$ cells led to the disappearance of the Fn14 puncta and its accumulation in perinuclear region in a TWK-independent manner.

To assess the role of autophagy in Fn14 trafficking from the vesicular structures we knocked down *BECN1* in ATG4B$^{DN}$ cells, which resulted in stronger accumulation of Fn14 (Fig. 5c). The sequence of events in Fn14 trafficking from endosomes was further determined by knocking down *TSG101* in ATG4B$^{DN}$ cells. While *TSG101* knockdown did not affect Fn14 degradation in WT HeLa cells (Fig. 2a), it led to vesicle expansion in autophagy-deficient cells (Fig. 5d). Taken together, these results suggest that Fn14 is first internalized to an endosomal compartment, from which it can be sorted to degradation in the lysosome by autophagy or the MVB machinery.

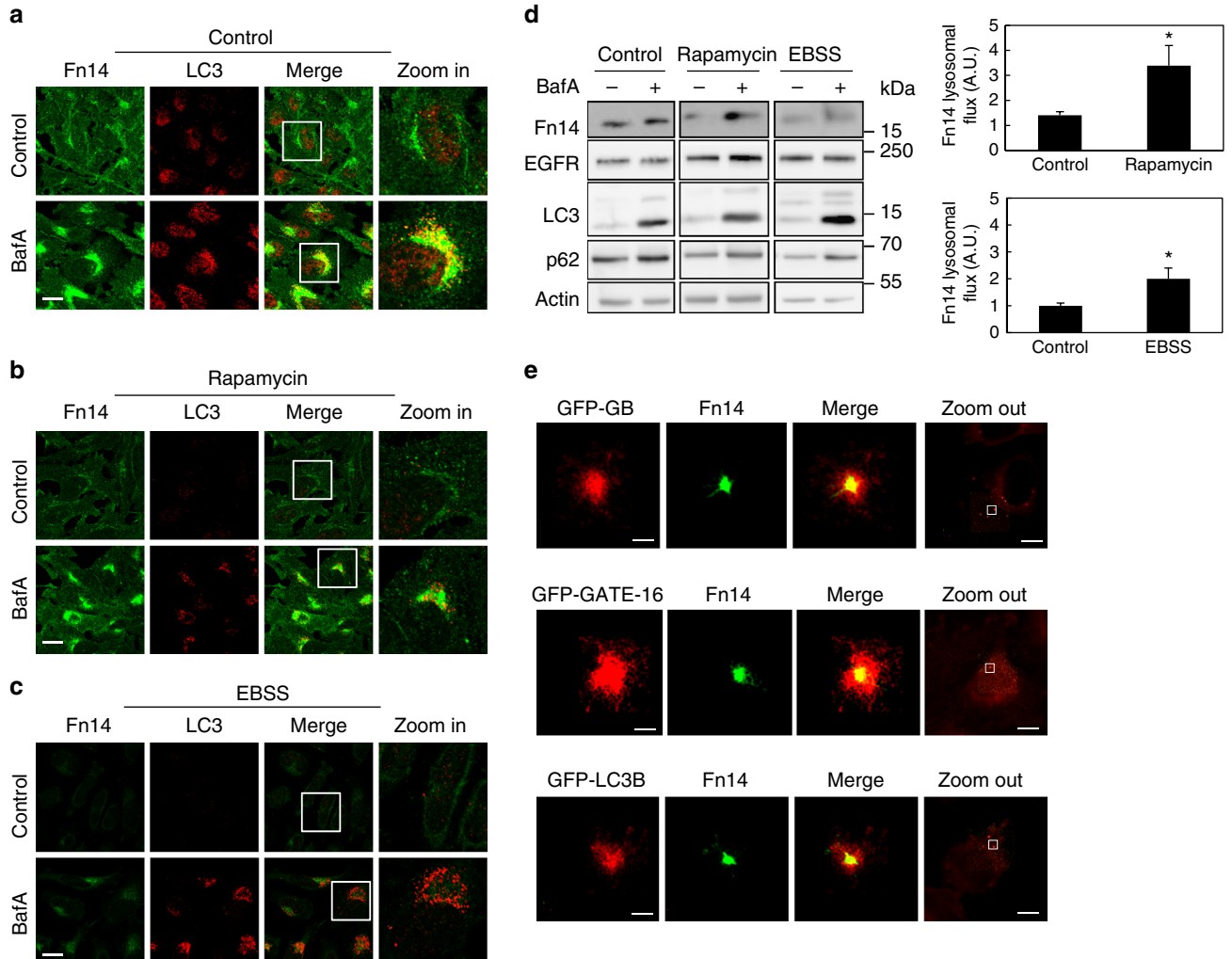

**Fig. 3** Induction of autophagy augments the degradation of Fn14. **a–c** HeLa cells were incubated for 12 h with BafA as indicated without further treatment (**a**) or with rapamycin (**b**) or starved in EBSS (**c**), immunostained for Fn14 and LC3, and analyzed by confocal microscopy (scale bar, 20 μm). Large magnification of stained cells is presented in the right column. **d** Proteins extracted from cells treated as in **a–c** were immunoblotted (left panel) and lysosomal flux was calculated (right panel) (*P < 0.05, n = 2 biological repeats performed in triplicates, comparisons by T test; mean ± s.e.m). **e** HeLa cells stably expressing GFP-GABARAP (GFP-GB), GFP-GATE-16, and GFP-LC3B were immunostained for GFP and Fn14 and analyzed by 3D STORM (scale bar, 300 nm). Landscape images are presented in the right column (scale bar 10 μm)

**Atg8 proteins differentially regulate Fn14.** Atg8 proteins have been implicated in selective recruitment of cargo to autophagosomes[21,22,24,28]. To determine whether individual Atg8 proteins have specific roles in Fn14 turnover, we knocked down each of the six Atg8 family members (GABARAP (GB), GABARAPL1 (GBL1), GATE-16, LC3A, LC3B, and LC3C) and revealed different phenotypes as follows (Fig. 6a and Supplementary Fig. 2f and 6a–c): knockdown of GABARAP or LC3C led to an increase in Fn14 levels, while knockdown of GATE-16 or LC3B resulted in accumulation of Fn14 in large vesicular structures (white arrowheads) without affecting its levels; moreover, higher Fn14 levels upon LC3C knockdown were unaffected by treatment with TWK, whereas the effect of LC3B knockdown on the accumulation of Fn14 in vesicles was mostly observed upon TWK treatment. We conclude that the regulatory role of the LC3 subfamily on Fn14 is weaker than that of the GABARAP subfamily. Notably, no effect on Fn14 was observed upon the knockdown GABARAPL1 or LC3A (Fig. 6a and Supplementary Fig. 6a,b).

To characterize the subcellular site in which Fn14 accumulates in response to GABARAP knockdown we co-stained Fn14 with ERGIC or Golgi markers in immunofluorescence. Fn14

accumulated mainly at the ERGIC and to a lesser extent at the Golgi complex (Fig. 6b and Supplementary Fig. 6d), yet as indicated by treatment with BafA the degradation of Fn14 was not completely abolished upon this knockdown (Fig. 6a and Supplementary Fig. 6a). A similar phenotype was also observed within the U251 glioblastoma cell line (Supplementary Fig. 6e). To directly test the effect of GABARAP knockdown (siGB) on Fn14 levels, we simultaneously transfected HeLa cells with siGB and an siRNA-resistant (silent mutant) GFP-GABARAP (GFP-GB). While expression of siRNA-resistant GFP-GB in siGB-knockdown cells showed a decrease in Fn14 levels compared with cells transfected with siGB alone, indicating a rescue of the siGB effect, a siRNA-resistant non-lipidated form of GABARAP (GFP-GB$^{G116A}$) was unable to provide this rescue, indicating that lipidation of GABARAP is required to proper regulation of Fn14 (Supplementary Fig. 6f).

To study the role of GATE-16 in the regulation of Fn14, endogenous Fn14 and endocytic or autophagic markers (Supplementary Fig. 7a), as well as with TRAF2 (Fig. 6c) were monitored in GATE-16-knockdown cells. The Fn14-positive vesicles observed upon GATE-16 knockdown were found positive for LC3, p62,

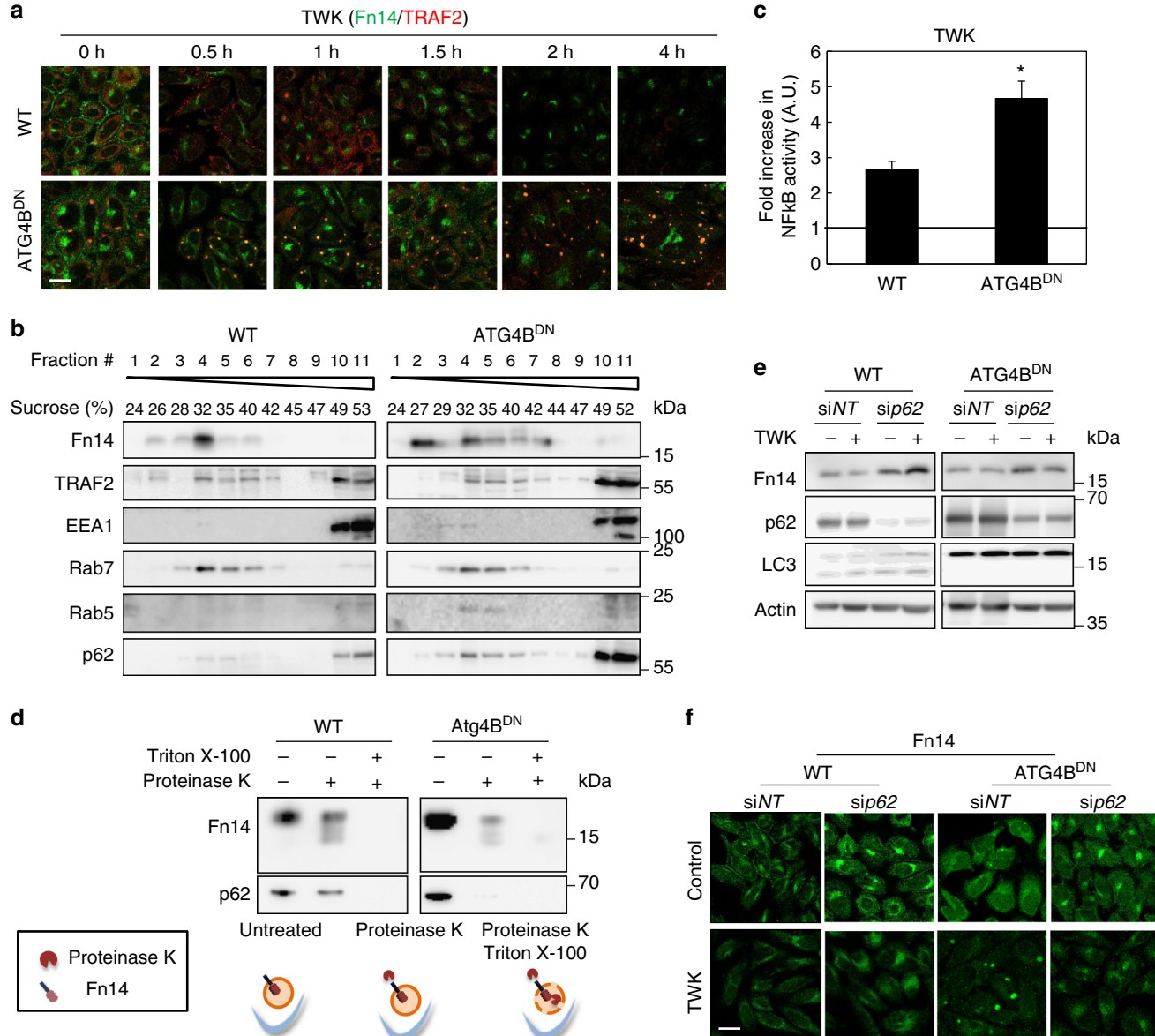

**Fig. 4** Fn14-positive vesicles accumulate in a TWEAK-dependent manner upon Atg8s inactivation by dominant-negative ATG4B. **a–f** WT or ATG4B[DN] stable expression HeLa cells were assayed as follows: **a** treated with TWK for indicated times, immunostained for Fn14 and TRAF2, and analyzed by confocal microscopy (scale bar, 20 μm); **b** treated with TWK and subjected to membrane floatation assay as detailed under Methods; **c** transfected with luciferase expression constructs as detailed under Methods, treated with TWK as indicated, and further subjected to luciferase assay as detailed to assess NF-kB activity (*$P < 0.05$, $n = 3$ biological repeats; comparisons by $T$ test; mean ± s.e.m.). The vertical line represents control in the absence of TWK; **d** subjected to Proteinase protection assay (upper panel) as detailed under Methods and schematically depicted (lower panel). **e, f** transfected with si*NT* or siRNA against p62 (si*p62*) and treated with TWK, extracted proteins were immunoblotted (**e**) or cells were immunostained for Fn14 and analyzed by confocal microscopy (scale bar, 20 μm) (**f**)

EEA1, and TRAF2, yet negative for ERGIC-53 (Supplementary Fig. 7c), indicating similarity to those vesicles observed in ATG4B[DN] cells. Similarly, vesicles detected upon knockdown of all Atg8s were negative for a Golgi marker GRASP65 (Supplementary Fig. 7c). Notably, knockdown of *ULK1*, a key autophagic factor acting at the initial stages of the process, in addition to *GATE-16* prevented vesicle formation (Supplementary Fig. 7e), suggesting that ULK1 facilitates formation of Fn14-positive vesicles upstream to their clearance by GATE-16. Interestingly, those vesicles were also positive for the Ras gene from rat brain GTPase-activating protein (RabGAP) TBC1D5 (Supplementary Fig. 7a), which plays a role in the endomembrane trafficking system and was previously shown to interact with Atg8 proteins

and to translocate from endosomes to autophagosomes[49–51]. *GABARAP* knockdown did not result in accumulation of Fn14 in endosomal compartments (Supplementary Fig. 7d), emphasizing the different roles of these Atg8 proteins. As for *GABARAP* above, *GATE-16* siRNA was also controlled by cotransfection with siRNA-resistant Myc-GATE-16. Staining either Fn14 or EEA1 upon GATE-16 knockdown in the presence of Myc-GATE-16 revealed a decrease in Fn14- and EEA1-positive vesicles (white arrow), compared with the knockdown alone (white arrow head) (Supplementary Fig. 7b). As for GABARAP above, this rescue of this phenotype was not demonstrated by the siRNA-resistant non-lipidated form of GATE-16 (Myc-GATE-16[G116A]) (Supplementary Fig. 7b). To further characterize the phenotypes of *GATE-16*

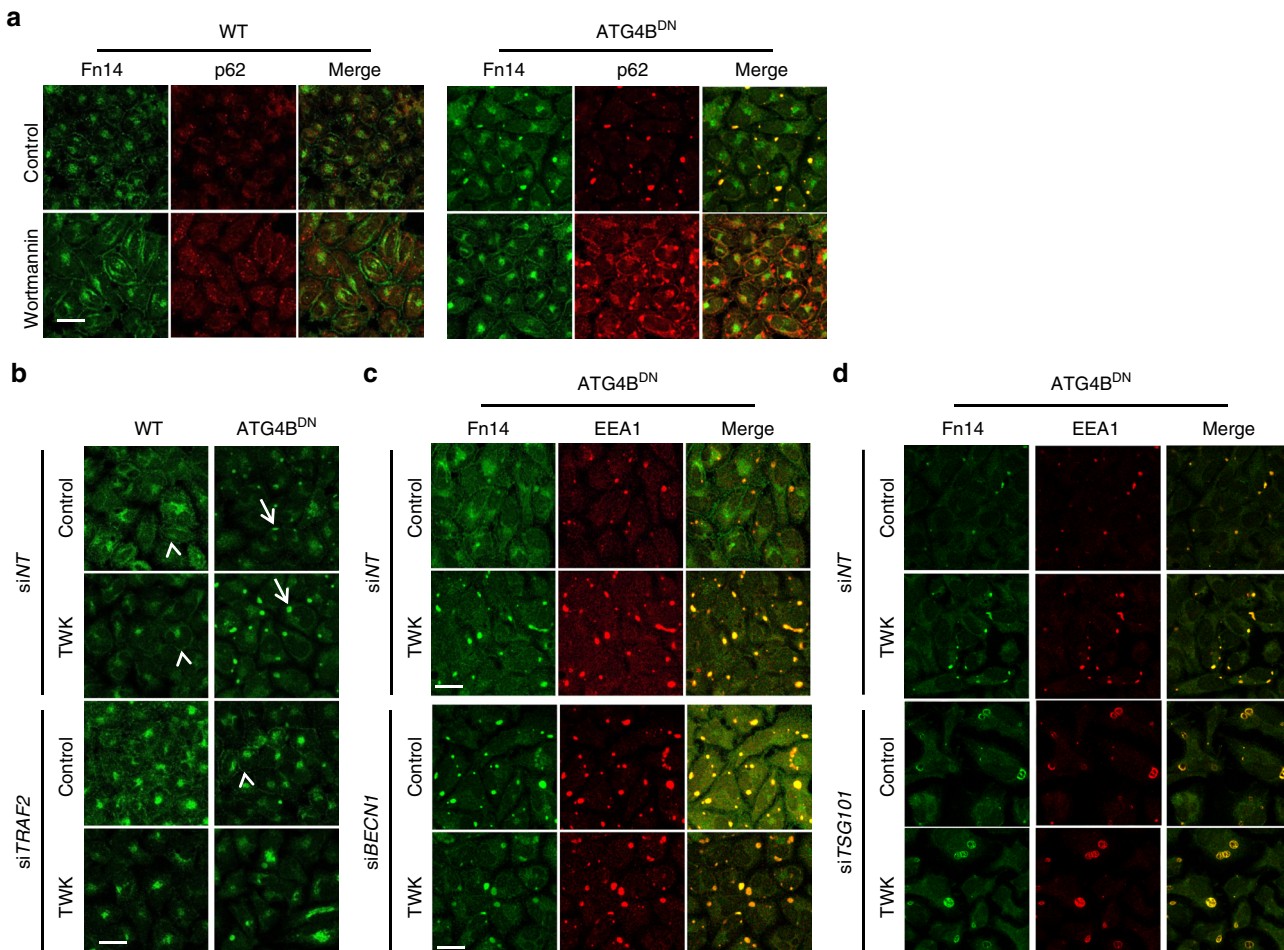

**Fig. 5** Autophagy and MVBs both contribute to Fn14 trafficking. **a–d** WT or ATG4B[DN] stable expression HeLa cells were treated as follows and analyzed by confocal microscopy (scale bar, 20 μm): **a** treated with wortmannin as indicated and immunostained for Fn14 and p62; **b** transfected with siNT or siRNA against TRAF2 (siTRAF2), treated with TWK as indicated, and immunostained for Fn14. Arrows and arrowheads point at Fn14 at vesicles and plasma membrane, respectively; **c**, **d** transfected with siNT or siRNA against *BECN1* (si*BECN1*) (**c**) or *TSG101* (si*TSG101*) (**d**), treated with TWK, and immunostained for Fn14 and EEA1

and *GABARAP* knockdown, we used 3D STORM. Apparently, knockdown of either *GATE-16* or *GABARAP* led to accumulation of Fn14 in the vicinity of LC3B-labeled open structures, suggesting that sequestration of Fn14 into phagophores is followed downstream by activities of GATE-16 or GABARAP (Fig. 6d).

Finally, we examined the effect of *GATE-16* knockdown on Fn14 signaling to NF-κB. Depletion of GATE-16 resulted in a significant increase of NF-κB activity, which was further elevated after treatment with TWK (Fig. 6e and Supplementary Fig. 7f). Evidently, overexpression of GATE-16 resulted in reduced NF-κB activity, both prior to and upon treatment with TWK (Supplementary Fig. 7g). Hence, in absence of GATE-16 an otherwise-degraded internalized Fn14 accumulates in intermediate structures, allowing ligand-independent recruitment of TRAF2 and signaling to NF-κB. On the other hand, the knockdown of *GABARAP* did not affect NF-κB activity (Fig. 6f and Supplementary Fig. 6g), suggesting that regulation of Fn14 by GABARAP occurs at an early stage, before it reaches the plasma membrane.

## Discussion

In this work we identified a role for the autophagic machinery in regulation of Fn14. In contrast to other TNF receptors, Fn14 is short-lived and highly inducible and its cellular levels are strictly regulated[1,19]. We hypothesized that in order to maintain low levels and avoid spontaneous signaling Fn14 is regulated primarily by autophagy. Our data identify two autophagic checkpoints during the life cycle of Fn14 (Fig. 7): whereas GABARAP regulates early stages through ERGIC, GATE-16 mediates degradation following internalization from the plasma membrane. We found that Fn14 vesicles formed upon depletion of GATE-16 are positive for the RabGAP protein TBC1D5, which was recently implicated in reprogramming of endosomal trafficking to autophagosomes under autophagic conditions[49–51].

Our data indicate that GABARAP-mediated autophagy controls the cellular level of Fn14. Inhibition of the autophagic machinery in general and depletion of GABARAP in particular leads to accumulation of Fn14 in the ERGIC, in consistence with recent implications of ERGIC in autophagosome formation[52–54]. Internalized plasma membrane receptors are typically targeted by the ESCRT machinery through MVBs for lysosomal degradation[39,40]. Here we describe a pathway whereby internalized receptors are delivered for lysosomal degradation by autophagy in a manner mediated mainly by GATE-16. Thus, far our data imply that Fn14 puncta formed upon inactivation of Atg8s by dominant-negative ATG4B or depletion of GATE-16 are positive not only for autophagic and endosomal markers but also for TRAF2, a signal transducer of several pathways, including NF-κB activation[55–58]. This raises the possibility that upon inhibition of

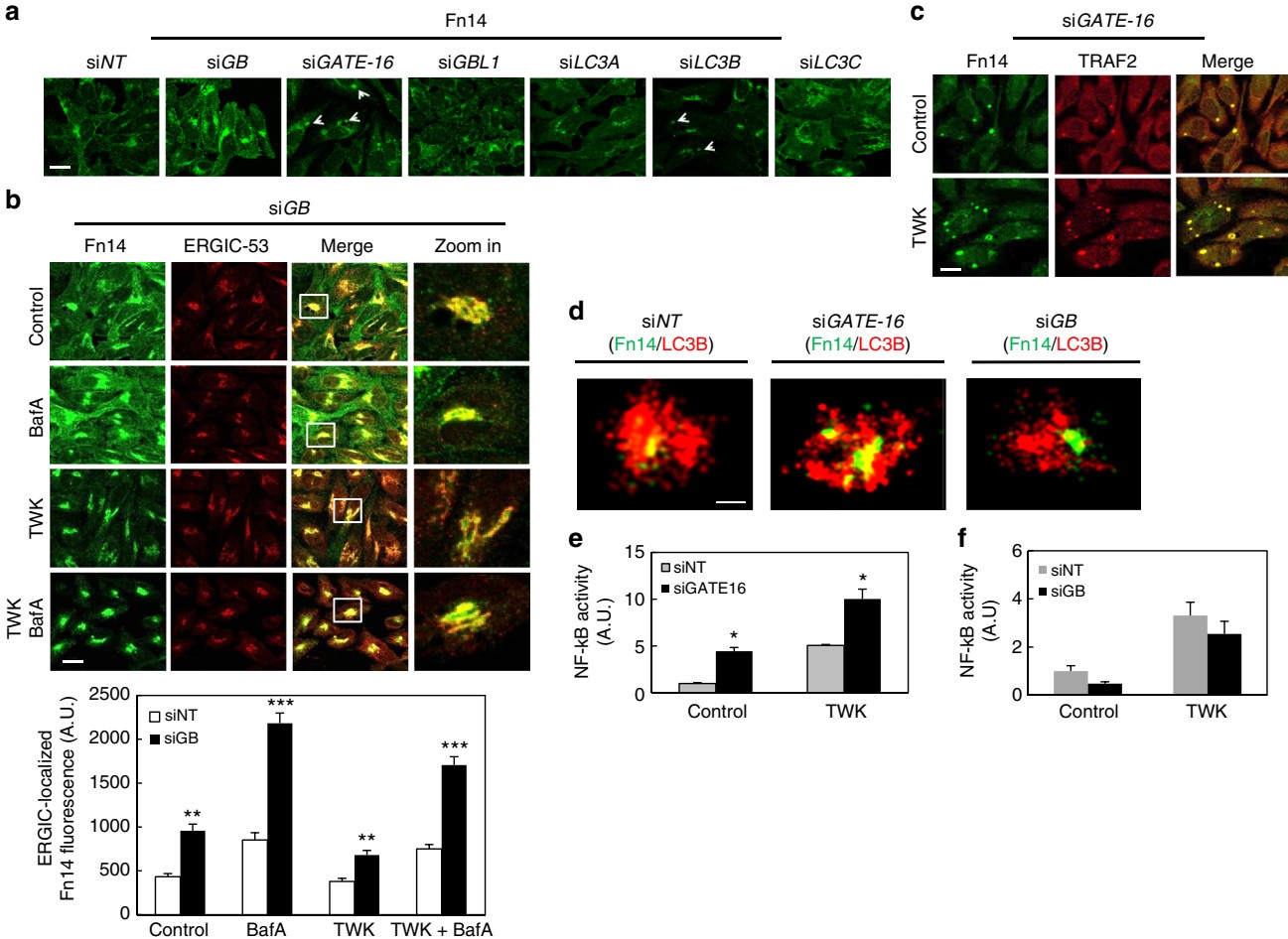

**Fig. 6** Atg8 proteins differentially regulate Fn14. **a** HeLa cells were transfected with si*NT* or siRNA against individual Atg8 isoforms as indicated, immunostained for Fn14, and analyzed by confocal microscopy (scale bar, 20 µm). **b** HeLa cells transfected with siRNA against GABARAP (si*GB*) were treated with TWK and BafA as indicated, immunostained for Fn14 and ERGIC-53, and analyzed by confocal microscopy (scale bar, 20 µm) (upper panel). Large magnification of stained cells is presented in the right column. ERGIC-localized Fn14 fluorescence was quantified from at least 10 fields (lower panel) (**$P < 0.0025$; ***$P < 0.0001$, comparisons by ANOVA. si*NT*—control, $n = 9$ cells; BafA, $n = 14$ cells; TWK, $n = 19$ cells; TWK baf, $n = 22$ cells. Si*GB*—control, $n = 23$ cells; BafA, $n = 27$ cells; TWK, $n = 32$ cells; TWK + BafA, $n = 28$ cells; mean ± s.e.m.). **c** HeLa cells transfected with siRNA against GATE-16 (si*GATE-16*) were treated with TWK as indicated, immunostained for Fn14 and TRAF2, and analyzed by confocal microscopy (scale bar, 20 µm). **d** HeLa cells transfected with si*NT*, si*GATE-16*, or si*GB* were immunostained for LC3B and Fn14, and analyzed by 3D STORM (scale bar, 300 nm). **e**, **f** Cells were transfected as with si*NT* or si*GATE-16* (**e**) or si*GB* (**f**) and after 24 h were transfected with luciferase expression constructs as detailed under Methods, treated with TWK, and further subjected to luciferase assay as detailed to assess NF-kB activity (*$P < 0.01$, comparisons by $T$ test; mean ± s.e.m.; $n = 3$ biological repeats and $n = 2$ biological repeats performed in triplicates, respectively)

autophagy Fn14 continues to signal by association with TRAF2 in the vicinity of phagophores.

The autophagic process may be tightly regulated and highly selective through cargo recognition and recruitment by its machinery, e.g., p62[22,24,28]. In this study we demonstrated that recruitment of Fn14 to the autophagic system from either ERGIC or early endosomes is highly p62-dependent. To our knowledge, this is the first reported sequestration of a plasma membrane receptor into autophagosomes by p62. The finding that GABARAP and GATE-16, despite their high structural similarity act differentially in this system, is rather surprising. We show here that TRAF2 is essential for Fn14 recruitment to GATE-16- but not GABARAP-dependent autophagy. As p62 is known to interact with poly- and mono-ubiquitinated proteins[22,59–61], it is important to further determine whether ubiquitination of Fn14 (or associated proteins) selectively targets Fn14 for autophagic degradation by different Atg8 family members. Future studies in this direction may clarify this important issue.

These pivotal findings clearly demonstrate involvement of the autophagic machinery in regulation of a TNF receptor, impinging on its trafficking as well as function, hence on its physiological roles. While the pathway unveiled in this study is not taken by other receptors such as TNFR1 or EGFR and therefore appears unique to Fn14, further investigation is warranted to determine the possibility of autophagic regulation on additional TNF receptors and other plasma membrane proteins. As Fn14 is known to regulate wound healing and repair[1,19], our findings may contribute new insights on the involvement of autophagy in this process. We speculate that when Fn14 is upregulated, as previously shown upon injury, autophagy is consequently induced in order to negatively feedback Fn14 signaling and maintain its homeostasis. To summarize, this study characterizes a regulatory role of the autophagic machinery on the TNF receptor Fn14, and establishes differential functions of distinct Atg8 factors in this process. This regulatory framework contributes to our knowledge of autophagic involvement in the

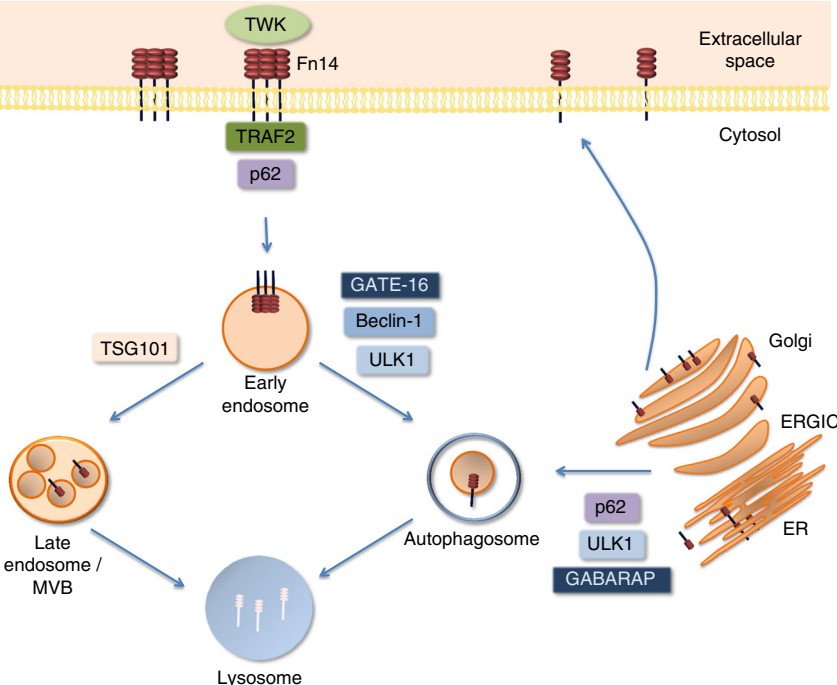

**Fig. 7** Regulation of Fn14 by the autophagic machinery. GABARAP-mediated autophagy is responsible for the main pool of the cellular Fn14 and in its absence the receptor accumulates in ERGIC. GATE-16-mediated autophagy and the canonical MVB pathway are responsible in parallel for degradation of Fn14 in endocytic compartments. In absence of GATE-16, Fn14-positive endosomes accumulate at the vicinity of open phagophores. Recruitment of Fn14 to either GABARAP- or GATE-16-dependent autophagosomes is mediated by the cargo receptor p62. Recruitment of internalized Fn14 through the GATE-16-dependent pathway requires TRAF2. The mechanism responsible for recruitment Fn14 to GABARAP-dependent autophagy is still unknown

endocytic pathway and to potential development of new therapeutic targets for Fn14 signaling.

## Methods

**Cell culture and treatment**. All cell lines were obtained from the Weizmann Institute Cell-Line Core and NCI-60. HeLa cells were grown on alpha minimum essential media, U251 cells on Dubeco's modified Eagle's medium, and Ovcar-8 cells on RPMI 1640. All cell cultures were supplemented with 10% fetal calf serum (FCS) at 37 °C in 5% $CO_2$. For induction of autophagy, cells were treated with 100 nM rapamycin (Sigma) or starved by washing twice with phosphate buffer saline (PBS) (Biological Industries) and incubating as above in EBSS medium (Biological Industries). Lysosomal degradation was inhibited by 0.1 μM BafA (LC Laboratories), 20 mM $NH_4Cl$, or 200 μM CQ (both from Sigma). Proteasomal degradation was inhibited by 0.25 μM Velcade. PI3P production was inhibited by 1 μM wortmannin (Sigma) or by 10 μM VPS34-IN2 (Novartis). Fn14, EGFR, and TNFR1 were activated by 100 ng/ml TWK (PeproTech), 10 ng/ml EGF (Sigma), and 50 ng/ml TNFα (Sigma), respectively. BafA and TWK were applied for 4 h prior to further treatment unless otherwise indicated, while other reagents were applied for indicated durations. The Atg8's CRISPR/Cas9 KO cells were kindly given by Dr. Lazarou lab. All cell lines were routinely inspected for mycoplasma contamination on a monthly basis.

**Antibodies**. Primary antibodies were diluted 1:1000 for western blotting (WB) and 1:200 for immunofluorescence (IF), unless stated otherwise. Mouse monoclonal anti-Fn14 (sc56250), mouse monoclonal anti-TBC1D5 (sc376296), rabbit polyclonal anti-TRAF2 (sc7187), rabbit polyclonal anti-TNFR1 (sc7895), rabbit polyclonal anti-p62 (sc25575), and mouse monoclonal anti-p62 (sc28359) were purchased from Santa-Cruz Biotechnology; mouse monoclonal anti-p62 was purchased from Abnova (H00008878, 1:3000 dilution for WB); rabbit polyclonal anti-Calnexin (ab22595), rabbit polyclonal anti-LAMP1 (ab24170), rabbit monoclonal anti-Rab7 (ab137029), and mouse monoclonal anti-Rab5 (ab66746) were purchased from Abcam; rabbit polyclonal anti-EEA1 (2411) and rabbit anti-EGFR (4267) were purchased from Cell Signaling; rabbit polyclonal anti-GATE-16 (PM038) and rabbit polyclonal anti-GABARAP (PM037) were purchased from MBL; rabbit polyclonal anti-ERGIC-53 (E1031), rabbit polyclonal anti-ATG3 (A3231), and rabbit polyclonal anti-ATG7 (A2856) were purchased from Sigma; mouse monoclonal anti-TSG101 (GTX70255) was purchased from Genetex; mouse monoclonal anti-Actin (69100, 1:5000 dilution) was purchased from Millipore; donkey anti-mouse Alexa488 (711545152), donkey anti-rabbit Cy5 (711175152), and donkey anti-rabbit Cy3 (711165152) were purchased from Jackson ImmunoResearch; rabbit polyclonal anti-GRASP65 kindly provided by Sima Lev; rabbit

polyclonal anti-WIPI1 was a gift from Tassula Proikas-Cezanne; rabbit polyclonal anti-p53 was a gift from Moshe Oren; rabbit polyclonal anti-LC3 antibody was produced by immunization of a rabbit with a peptide of the first 14 amino acids (excluding Met1) of human LC3B with an additional cysteine (PSEKTFKQRRTFEQC).

**Transfection**. The following siRNA SMARTpool sequences, each consisting of four RNA duplexes at a final concentration of 50 nM, were used. A total siRNA concentration of 150 nM was used for the knockdown of an entire Atg8 subfamily. siRNA oligos (Dharmacon or Sigma) were based on the same sequences; control, non-targeting (# D-001206-14), *p62* (# M-010230-00), *LC3A* (# M-013579-00), *LC3B* (# M-012846-00), *LC3C* (# M-032399-01), *GATE-16* (# M-006853-02), *GABARAP* (# M-012368-01), *GABARAPL1* (# M-014715-01), *ULK1* (# M-005049-00), *BECN1* (# M-010552-01), *TRAF2* (# M-005198-00), *ATG3* (# L-015375-00), and *ATG7* (# M-020112-01). Of note, we knocked down both *ATG3* and *ATG7* since either alone did not sufficiently impair their activity in our system, most likely due to the residual activity of these enzymes. si*TSG101* was obtained from Sigma (5′-CGAUGGCAGUUCCAGGGAA-3′). si*Fn14* was purchased from Santa Cruz (sc-43764). siRNA was transfected into cells using DharmaFECT 1 transfection reagent 72 h prior to further treatment as indicated.

Plasmids were transfected using jetPEI transfection reagent (Polyplus transfection) according to the manufacturer's instructions 48 h prior to further treatment as indicated. pGFP–C1-GABARAP, pGFP–C1-GATE-16, ATG4B^DN, pGFP–C1-GATE (G116A), and pGFP–C1-GABARAP (G116A) were produced in our lab. siRNA-resistant GFP-GABARAP and GFP-GABARAP (G116A) were constructed by introduction of silent mutations in all siRNA-targeted regions. Flag-Fn14 and pcDNA-TNFR1 were kindly provided by David Wallach.

**Stable expression of ATG4B^DN**. Human ATG4B^DN was cloned from pcDNA3 and into a pLEX_TRC 206 plasmid. The lentiviral packaging vectors pCMV-VSV-G, psPAX2, and pLEX_TRC 206 were transfected into HEK293 cells, viral particles were harvested from cell lysates and culture media, and filtrated by Steriflip (Merck). Recombinant virii were used to infect JW HeLa cells. Target cells were plated in six-well plates at a density of $1 \times 10^6$ cells/ml, infected the following day by recombinant lentivirus, and further incubated for 48 h. Cells were then washed three times in PBS, supplemented with fresh media, and underwent selection in 10 μg/ml Blasticidin for a week for stable expression of ATG4B^DN, before inhibition of autophagy was finally confirmed by immunoblotting.

**Protein extraction**. Cells were lysed in RIPA buffer (0.1 M NaCl, 5 mM EDTA, 0.1 M sodium phosphate, pH 7.5, 1% (v/v) Triton X-100, 0.5% (w/v) sodium

deoxycholate, and 0.1% (w/v) sodium dodecyl sulfate) supplemented with protease inhibitors (Calbiochem), lysates were centrifuged (5 min, 700 × g, 4 °C), and protein concentration was determined by Bio-Rad Protein Assay (Bio-Rad). Equal amounts of proteins (30 µg) were boiled in sample buffer (5 min, 95 °C) and immunoblotted.

**Immunoblotting**. Proteins were separated by 12% SDS–polyacrylamide gel electrophoresis and transferred to a polyvinylidene difluoride membrane. The membrane was blocked (5% (w/v) milk in PBS, 1 h, room temperature (RT)), incubated with primary antibody (overnight, 4 °C), washed (0.1% (v/v) Tween-20 in PBS, ×3), incubated with secondary antibody (1 h, RT), and washed (as above). Finally, the membrane was treated with an ECL reagent (Biological Industries) and observed under chemiluminescence (ImageQuant LAS 4000, Fuji). Uncropped western blots for all main figures are presented in the Supplementary Information. We indicate where reblotting took place.

**Immunostaining**. Cells cultured on sterile coverslips (13 mm) and treated as indicated were fixed and permeabilized (100% methanol, 10 min, −20 °C), washed (PBS, ×3), blocked (10% FCS in PBS, 30 min, RT), and incubated with primary antibody (1 h, RT), washed (as above) and incubated with secondary antibody (30 min, RT).

**Confocal fluorescence microscopy**. Immunostained cells were observed under a confocal laser scanning microscope (FV1000, UPLSAPO 60 × 1.35 numerical aperture (NA) oil immersion, Olympus) and images were analyzed by Fluoview software (Olympus). Colocalization between the different markers was analyzed from at least two individual experiments by Imaris 8.2.0 (Bitplane). Mean of intensity of Fn14 in the ERGIC and Golgi compartment was analyzed by ImageJ (NIH).

**3D STORM**. Cells cultured on coverslips attached to Petri dishes (MatTek P35G-1.5-14-C), treated as indicated, and immunostained were observed under Vutara SR-200 microscope (Bruker), using ×60, 1.2 NA water-immersion objective (Olympus) and an Evolve 512 EMCCD camera (Photometrics) with gain set at 50 and frame rate at 50 Hz. Dual color measurements were performed using Cy5 and Alexa568 fluorophores with a collection of 4000 and 6000 frames, respectively, for each channel. Maximal excitation was 6 kW/cm$^2$ for the 647 nm laser, and 5 kW/cm$^2$ for the 561 nm laser. The power of 405 nm activation laser was gradually increased to a maximum of 0.05 kW/cm$^2$. Imaging was performed in the presence of imaging buffer (7 µM glucose oxidase, 56 nM catalase, 2 mM cysteamine (all from Sigma), 50 mM Tris, pH 8, 10 mM NaCl, and 10% glucose). Data were analyzed by Vutara SRX 6.01.12 software.

**Real-time PCR analysis**. Total RNA was isolated from cells (NucleoSpin RNA II kit, Macherey-Nagel), and RNA (1 µg) was reverse-transcribed (M-MLV reverse transcriptase, Promega) using random hexamer primers (Amersham). Real-time PCR was performed using Platinum® SYBR® Green (Invitrogen) on an ABI 7300 instrument (Applied Biosystems). A20 mRNA levels measured from cDNA (forward primer, 5′-TACCCTTGGTGACCCTGAAG-3′; reverse primer, 5′-AATC TTCCCCGGTCTCTGTT-3′) were normalized to HPRT housekeeping control (forward primer, 5′-GCTGGCAACTGGAGTCTCTC-3′; reverse primer, 5′-TTGT CCCATTCATCATTCCA-3′).

**NF-kB activity luciferase assay**. Luciferase assay was performed with a kit (Promega) according to the manufacturer's instructions. Briefly, cells were cotransfected with a plasmid (250 ng) containing luciferase reporter driven by HIV promoter to report NF-kB activity and a plasmid (50 ng) containing Renilla luciferase driven by RSV promoter for normalization. Transfections were done in triplicates 48 h before indicated treatments. After treatments cells were harvested with luciferase lysis buffer, and the lysate was split into triplicates for reporter luciferase assay and triplicates for Renilla luciferase assay—each with its respective substrate. Finally, luminescence was read with a luminomiter (Monolight 2010, Analytical Luminescence Laboratory). The average readouts of luciferase reporter were normalized to average readouts of Renilla luciferase and the result was presented as NF-kB activity.

**Membrane flotation assay**. Cells were homogenized with the Balch homogenizer, homogenates (2 mg protein) were adjusted to 2 M sucrose, placed at the bottom of rotor tubes, overlaid with 1.75, 1.5, 1.25, 1, and 0.75 M sucrose, and centrifuged 102 000 × g (slow acceleration and deceleration) overnight, 4 °C. Fractions from the top of the gradient were collected, and sucrose densities were calculated from their refractive indices. Proteins from each fraction (360 µl) were precipitated by 10% trichloroacetic acid (TCA), boiled in sample buffer (as above), and immunoblotted.

**Proteinase protection assay**. Cells cultured and treated in 15 cm dishes were detached by trypsin, washed (PBS, ×3), resuspended in 4 pellet volumes of homogenization buffer (10 mM Tris, pH 7.4, and 0.25 M sucrose) supplemented by protease inhibitor, and homogenized on ice with a Balch homogenizer. Unbroken cells and nuclei were removed (700 × g, 5 min, 4 °C) and equal amounts of protein

were centrifuged using a TLA 120.2 rotor (90 000 rpm, 30 min, 4 °C) for cytosol and membrane fractions, and the pellets were resuspended in homogenization buffer. Each fraction was then divided into equal volumes for treatments (30 min, 37 °C) of Proteinase K (10 µg/ml, Merck), Triton X-100 (0.4% (v/v)), or both—as indicated. Treatments were terminated by the addition of phenylmethylsulfonul fluoride (Sigma) (200 mM, 10 min on ice), proteins were precipitated by 10% TCA, boiled in sample buffer (as above), and immunoblotted.

**Quantitative and statistical analyses**. Large vesicles observed under confocal fluorescence microscopy were manually counted in six different fields per sample. Protein levels were quantified by Image Studio Lite software (LI-COR) and normalized to corresponding Actin levels. Lysosomal flux was calculated by dividing normalized protein level in BafA-treated samples by that of matching untreated sample. Values in figures—protein and mRNA levels, luciferase activity, and lysosomal flux—represent the mean value of at least three independent trials, error bars are standard error of mean. Statistical analyses were carried out using InStat 3 software package (GraphPad Software). Unless stated otherwise, significance of differences between different conditions was determined by T test.

## Data availability

All relevant data are available by request from the corresponding author.

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

## Acknowledgements

Z.E. is an incumbent of the Harold Korda Chair of Biology. We are grateful for funding from the Israel Science Foundation ISF (Grant #1247/15), the Legacy Heritage Fund (Grant #1935/16), and the Weizmann Institute Minerva center. We especially thank Dr. Michael Lazarou from Monash University for the Atg8s-knockout cells. We wish to thank Vladimir Kiss and Dr. Reinat Nevo for their assistance with confocal microscopy. We also thank Oren Shatz for his critical reading and insightful comments.

## Author contributions

H.W., M.F. and Z.E. conceived the study, designed and interpreted experiments, and wrote the manuscript. H.W., M.F., A.A., B.-C.T.-Y. and T.D. performed experiments and analyzed results. All authors commented on the manuscript.

## Additional information

**Competing interests:** The authors declare no competing interests

