## [Peer Review File · Nature Communications]

Reviewers' comments:

Reviewer #1 (Remarks to the Author):

In this manuscript, the authors sought to identify key factors that may mediate regulatory/degradation mechanisms of Fn14. Given that the molecular mechanisms that ultimately decide how Fn14 is regulated are poorly understood, the work from this paper could significantly contribute to a portion of the Fn14 literature that is substantially lacking. As a result, a new link between Fn14 regulation and autophagy has been proposed. Despite this novelty, the data presented are questionable due to a large amount of immunofluorescence where, perhaps, other experimental procedures could have been used. Concerns are outlined below.

1. Overall, the manuscript contained various grammatical errors which could not be overlooked in order for publication. Example: First sentence of the results section should read: 'To identify the mechanism that governs Fn14 turnover...' OR 'To identify a mechanism that governs Fn14 turnover...'
2. Many different proteins were used in experiments throughout the study, but the manuscript did not provide significant background within the introduction. It is therefore difficult to follow the meaning behind the experiments provided (Ex: GATE-16 and its role as an Atg8 family member, and TSG101 (MVBs and distinction between autophagy).
3. Figure 1. Fig 1A. The histogram below the western compare control and TWK treatment is vague and unclear. Clarification using the term 'flux' is needed because autophagic flux refers to the completion of the autophagic process, but here it seems that the authors are using flux to describe Fn14 accumulation as a result of lysosomal inhibition by BafA. In panel B-D, staining for Fn14 in these panels appears to be clustered or perinuclear, where a membrane-localized staining is expected, especially in the control group. A DAPI stain should be included in order to obtain a better idea of the localization.
4. Figure 2. For Figure 2A, the Fn14 blot is too dirty and it is difficult to interpret the data. For 2B, it is unclear why Flag-tagged Fn14 overexpressed? Using endogenous Fn14 in the parental MEF cells to show modulation of protein levels would add more value with Atg5^{-/-} cells. In Figure 2C, knockdown of ATG7 by siRNA is not convincing. Moreover, Fn14 levels are too low in the siNT group, making it hard to assess the modulation of Fn14 protein level. Why use a mixture of Atg3 and Atg7 siRNA? In Figure 2D-F is the point of this experiment to show Fn14 localizes to lysosomes or autophagosomes? It is hard to determine whether these punta structures are lysosomes or autophagosomes. Therefore, co-staining with LAMP-1 is recommended to distinguish between these two processes. Additionally, for the ATG4DN construct, ATG4BDN was used in the text. Please maintain consistency with nomenclature.
5. Figure 3. In 3A-C, it is unclear why LC3 staining appears predominantly nuclear? Upon induction or inhibition of autophagy, staining of LC3 should reveal the formation of puncta (dots), but this is not apparent in this experiment. Fn14 staining from panel C is hardly visible.
6. Figure 4. Fn14 should colocalized with TRAF2 upon TWEAK treatment, which is not obvious in Figure 4A. These images are too small to assess the localization of Fn14 and TRAF2 in autophagosomes.

7. Figure 6. In 6A, a western blot should be performed to validate any modulation of Fn14 protein levels using all of the siRNAs for the Atg8 proteins and the quantification of Fn14 in the puncta structure should be included. In addition, in 6A, there appears to be an increase in Fn14 in GB knockdown. However, in 6F, there was no increase in NF- κ B activity, which contradicts the finding that increased Fn14 protein expression results in NF κ B activation. Similarly, with GATE16 siRNA, it was observed that Fn14 accumulates with LC3, p62 and EEA1 (Supplementary 6A), which indicates that Fn14 is localized in autophagosomes and therefore unable to signal via NF κ B. However, with the luciferase data in 6E, increased NF- κ B activity is observed.
8. The study should examine other closely related members of the TNFR superfamily to Fn14 such as BCMA, BAFFR, EDAR, and TROY in addition to TNFR1. Are these closely related members to Fn14 protein stability regulated by autophagosome mediated degradation or is this process unique only to Fn14. This would strengthen the novelty of the study.

Reviewer #2 (Remarks to the Author):

This manuscript presents very interesting and current data about the role of the TNF receptor, and the distinct roles of the Atg8 family. The conclusions are interesting and important for the labs studying both Fn14 and autophagy. The final model opens up a new way of thinking about the role of selective autophagy and trafficking through the cell, and the Atg8 family.

The major issue with this manuscript is that many controls are missing and in some cases the quality of the data is poor and not convincing. This should be easily rectified and will substantially strengthen the manuscript.

Major points

1. Figure S1a. The experiment needs a positive control for Velcade activity as it had no effect
2. Figure 2. 2a) The experiment needs to include a western blot to show efficient knock-down of Tsg101. Note: In Figure S2a the panel is labelled "siFn14" when as I understand it is the antibody to Fn14 which is shown. Furthermore this blot is not convincing. It is difficult to see the BAFA effect.
3. What is TNFR1 and why was this used as a control in Fig. S2b?
4. Again in Figure 2d-e need controls for siRNAs of Atg8s.
5. Figure 3b and c- EBSS is very effective for short periods but in panel c the authors use 12 hrs. This should be repeated with shorter times to eliminate off-target effects of long-term treatment. The nuclear background with the LC3 antibody varies a lot- is there a reason for this?
6. Figure 3e. Under what conditions is this experiment done?
7. Figure 4d is an important experiment to show the accumulation of Fn14 on either autophagic or endocytic membranes. Unfortunately the data is not convincing and insufficient quality. In particular the ATG4B panel shows uneven loading of EEA1, smearing of the lanes. It is impossible to judge where the peak and co-fractionation occurs. In addition what do the authors conclude about the LC3 panel in this condition? This must be improved and repeated at least n=3.

8. Page 8, the authors must consider that wortmannin will also inhibit the endocytic pathway. Is FN14 internalized normally in the presence of wortmannin?
9. In Figure 6 the authors should show EEA1 distribution with siRNA GABARAP to show the lack of staining and to control again for internalization defects. And visa versa with siRNA GATE16 should be labelled with ERGIC antibodies.

Minor points

1. Abstract last sentence "autophagosome"
2. In Fig. 4c the legend says WIPI2. Can the authors clarify this?

Reviewer #3 (Remarks to the Author):

The TNF receptor Fn14 is a short-lived protein. The author found that autophagy contributes to the degradation of Fn14 through a selective autophagy process dependent on P62. Interestingly, the authors found that the ATG8 family proteins play differential roles in regulating the function and turnover of Fn14. Overall, the discovery is important and data are supportive. Specific points below:

Major points:

1. An important structure described in this work is the accumulation of Fn14 into big punctate compartments in the absence of all ATG8s or autophagy inhibition. It is important to further characterize this structure to clarify it is a preautophagosomal structure. For example, a thin section EM would help, and a proteinase K treatment to determine if Fn is sequestered into the autophagosome. Also it is important to clarify that whether the structure formed under different KDs are essentially the same or they are different.
2. Although the work suggests that the requirement of ATG8 lipidation in the regulation of Fn14, it is helpful to further clarify if the lipidation is required by rescue the siRNA treated cells with the lipidation deficient ATG8s.
3. The author will consider modify the model in the last Figure because 1. GATE16 is not responsible for the level of Fn14. Therefore it is not consistent in Fig.7 showing that the GATE16 pathway also directs to the lysosome. 2. The overall data is not enough to support two independent pathways controlling Fn14 trafficking because KD of all ATG8s leads to the similar localization of Fn14 with GATE16 alone. This indicates that ERGIC and endocytosis may function in a same pathway but have a relationship of up and down stream.
4. To further substantiate the data, the author will consider also perform a double labeling of Fn14 with other ATG8s with available antibodies.
5. The author will consider further discuss the possible scenario about why the ATG8 homologues are different in regulating the Fn14.

Minor points:

1. Does the perinuclear puncta of Fn14 in ATG8s all KD colocalize with the ERGIC or GRASP65?
2. In Fig.3e and 6d, why the LC3 vesicles are not hollow under the STORM microscope?
3. There is a typo in Fig.4C. WIPI1 should be WIPI2 according to the text.

4. Does KD of GABARAP or LC3C affect NF- κ B signal?

5. Fig 5a, the author should use other VPS34 specific inhibitors as well as KD ATG14 to confirm.

We thank the reviewers for supporting the novelty of our findings as well as for their constructive comments, which helped us improve the overall quality of our manuscript. As detailed below we addressed all the referees' comments. In the revised manuscript we added the missing controls and experiments that helped clarify our working model. Our modified model implies that GABARAP-mediated autophagy is responsible for Fn14 cellular levels and in its absence an inactive receptor (which does not signal to NF- κ B) accumulates in the ERGIC complex. GATE-16-mediated autophagy, in parallel to the canonical MVB pathway, is responsible for the degradation of Fn14 localized in endocytic compartments. In the absence of GATE-16, Fn14-positive endosomes accumulate at the vicinity of autophagic membranes while maintaining their NF- κ B signaling activity. In the discussion section of the revised manuscript we propose a working hypothesis for the differential roles of the different Atg8s discovered in this study.

Reviewers' comments:

Reviewer #1 (Remarks to the Author):

In this manuscript, the authors sought to identify key factors that may mediate regulatory/degradation mechanisms of Fn14. Given that the molecular mechanisms that ultimately decide how Fn14 is regulated are poorly understood, the work from this paper could significantly contribute to a portion of the Fn14 literature that is substantially lacking. As a result, a new link between Fn14 regulation and autophagy has been proposed. Despite this novelty, the data presented are questionable due to a large amount of immunofluorescence where, perhaps, other experimental procedures could have been used. Concerns are outlined below.

1. Overall, the manuscript contained various grammatical errors, which could not be overlooked in order for publication. Example: First sentence of the results section should read: 'To identify the mechanism that governs Fn14 turnover...' OR 'To identify a mechanism that governs Fn14 turnover...'

We thank the reviewer for pointing out this issue. The revised manuscript went through careful English editing, which helped clarify our messages.

2. Many different proteins were used in experiments throughout the study, but the manuscript did not provide significant background within the introduction. It is therefore difficult to follow the meaning behind the experiments provided (Ex: GATE-16 and its role as an Atg8 family member, and TSG101 (MVBs and distinction between autophagy)).

We added to the text of the revised manuscript short descriptions for each of the factors used in this study.

3. Figure 1. Fig 1A. The histogram below the western compare control and TWK treatment is vague and unclear. Clarification using the term ‘flux’ is needed because autophagic flux refers to the completion of the autophagic process, but here it seems that the authors are using flux to describe Fn14 accumulation as a result of lysosomal inhibition by BafA.

We thank the reviewer for this comment. To clarify this issue, we modified the graph legend to Fn14 degradation levels and added a more detailed explanation to the Materials and Methods section of the revised manuscript.

In panel B-D, staining for Fn14 in these panels appears to be clustered or perinuclear, where a membrane-localized staining is expected, especially in the control group. A DAPI stain should be included in order to obtain a better idea of the localization.

We thank the reviewer for raising this important issue. In the revised manuscript we present more representative images alongside DIC images to clarify the cellular landscape.

4. Figure 2. For Figure 2A, the Fn14 blot is too dirty and it is difficult to interpret the data.

We now replaced the western blot with a clearer result in Figure 2A and Supplementary Figure 2B of the revised manuscript. While we are aware of the poor quality of Fn14 western blots presented throughout this study, we hope the reviewer may appreciate the difficulty in working with this poorly expressed endogenous membrane receptor.

For 2B, it is unclear why Flag-tagged Fn14 overexpressed? Using endogenous Fn14 in the parental MEF cells to show modulation of protein levels would add more value with Atg5^{-/-} cells.

We used ectopic expression of Fn14 in this system as our anti-Fn14 antibody failed to detect the endogenous protein in these cells. To directly address the reviewer's comment, we now also demonstrate the accumulation of endogenous Fn14 in autophagy-deficient HeLa cells, knocked out of all six Atg8 proteins (Figure 2g and Supplementary Figure S2f) (Nguyen et al. J Cell Biol. 2016 Dec 19;215(6):857-874).

In Figure 2C, knockdown of ATG7 by siRNA is not convincing. Moreover, Fn14 levels are too low in the siNT group, making it hard to assess the modulation of Fn14 protein level. Why use a mixture of Atg3 and Atg7 siRNA?

To address this comment we use higher exposure of the ATG7 western blot. In addition, it is well recognized among members of the autophagic community that knockdown of some of the key autophagic factors (especially enzyme) is not always sufficient to block autophagy. As both ATG7 and ATG3 are enzymes, we found that the knockdown of either did not affect autophagy. We therefore present data in which both enzymes were knocked down, leading to inhibition of autophagic flux. We added a clarification concerning this issue to the Materials and Methods section of the revised manuscript.

In Figure 2D-F is the point of this experiment to show Fn14 localizes to lysosomes or autophagosomes? It is hard to determine whether these puncta structures are lysosomes or autophagosomes. Therefore, co-staining with LAMP-1 is recommended to distinguish between these two processes. We thank the referee for this comment; co-staining with LAMP1 was accordingly added to Supplementary Figure 2i,k of the revised manuscript.

Additionally, for the ATG4DN construct, ATG4BDN was used in the text. Please maintain consistency with nomenclature.

This inconsistency was corrected throughout the revised manuscript.

5. Figure 3. In 3A-C, it is unclear why LC3 staining appears predominantly nuclear? Upon induction or inhibition of autophagy, staining of LC3 should reveal the formation of puncta (dots), but this is not apparent in this

experiment. Fn14 staining from panel C is hardly visible. Appearance of LC3 labeling in the nucleus in resting cells is common (although not always consistent) and usually appears when no or only few autophagosomes are visible. We consider this a background staining. To clarify autophagosome labeling by LC3, in the revised manuscript we present data in which the background was equally reduced in all images.

6. Figure 4. Fn14 should colocalized with TRAF2 upon TWEAK treatment, which is not obvious in Figure 4A. These images are too small to assess the localization of Fn14 and TRAF2 in autophagosomes. In the revised manuscript we present enlarged images clarifying the colocalization of TRAF2 and Fn14 in Atg4BDN cells. As indicated in the main text of the revised manuscript, Fn14 and TRAF2 in WT HeLa cells are rapidly degraded following TWEAK treatment (and therefore are not detectable) however; the colocalization of these proteins is evident in the presence of Atg4BDN mutant.

7. Figure 6. In 6A, a western blot should be performed to validate any modulation of Fn14 protein levels using all of the siRNAs for the Atg8 proteins and the quantification of Fn14 in the puncta structure should be included. The western blots monitoring Fn14 levels upon knockdown of individual Atg8 are presented in Supplementary Figure 6a,b. The knockdown of all Atg8s is shown in Supplementary Figure 2g. Quantification of Fn14 levels in the puncta formed upon knockdown of GATE-16 or LC3B was added to Supplementary Figure 6c.

In addition, in 6A, there appears to be an increase in Fn14 in GB knockdown. However, in 6F, there was no increase in NF-kB activity, which contradicts the finding that increased Fn14 protein expression results in NFkB activation. Similarly, with GATE16 siRNA, it was observed that Fn14 accumulates with LC3, p62 and EEA1 (Supplementary 6A), which indicates that Fn14 is localized in autophagosomes and therefore unable to signal via NFkB. However, with the luciferase data in 6E, increased NF-kB activity is observed. Our working model implies that GABARAP-mediated autophagy is responsible for degradation of ERGIC-localized Fn14 prior to its arrival to the plasma membrane. Based on this we predict that the accumulated receptor observed in absence of GABARAP remains unfunctional and should not affect NF-kB

signaling. GATE-16 on the other hand is mainly responsible for the degradation of endosome-localized Fn14 following internalization, which would otherwise engage in extended NF- κ B signaling. As this might not have been clear in the original manuscript, in the revised manuscript we modified our schematic model and the accompanying text to clarify this issue.

8. The study should examine other closely related members of the TNFR superfamily to Fn14 such as BCMA, BAFFR, EDAR, and TROY in addition to TNFR1. Are these closely related members to Fn14 protein stability regulated by autophagosome mediated degradation or is this process unique only to Fn14. This would strengthen the novelty of the study.

This reviewer's comment is well appreciated, however in our study we focused on Fn14 as a very short-lived protein degraded by the lysosome. This by itself required a long period of calibration and fine-tuning to follow Fn14 at its endogenous level. The well-characterized TNFR1 and EGFR were additionally examined as negative controls to the effect of autophagy in our system. The characterization of other TNF-like receptors would likely entail extensive research beyond the scope here. We provide a new and rather surprising interface between the pathways of autophagy and Fn14, and believe that future studies by our group and others will determine its broader applicability. Accordingly, a statement of clarification was added to the Discussion section of the revised manuscript.

Reviewer #2 (Remarks to the Author):

This manuscript presents very interesting and current data about the role of the TNF receptor, and the distinct roles of the Atg8 family. The conclusions are interesting and important for the labs studying both Fn14 and autophagy. The final model opens up a new way of thinking about the role of selective autophagy and trafficking through the cell, and the Atg8 family. The major issue with this manuscript is that many controls are missing and in some cases the quality of the data is poor and not convincing. This should be easily rectified and will substantially strengthen the manuscript.

Major points

1. Figure S1a. The experiment needs a positive control for Velcade activity as it had no effect

A p53 control for the effect of Velcade was added to Supplementary Figure 1a of the revised manuscript.

2. Figure 2. 2a) The experiment needs to include a western blot to show efficient knock-down of Tsg101. Note: In Figure S2a the panel is labelled “siFn14” when as I understand it is the antibody to Fn14 which is shown. Furthermore this blot is not convincing. It is difficult to see the BAFA effect. All remarks to Figures 2 and Supplementary Figure 2 were addressed in the revised manuscript. The western blot of Figure 2a was replaced with a clearer blot.

3. What is TNFR1 and why was this used as a control in Fig. S2b? TNFR1 – TNF Receptor 1 – is a well-characterized member of the TNF receptors family to which Fn14 belongs. We used it in our experiments as a negative control to show the unique regulation of Fn14 by autophagy. A clarification statement was added to the revised manuscript.

4. Again in Figure 2d-e need controls for siRNAs of Atg8s. The western blot showing the effect of siAtg8s is in Supplementary Figure 2g.

5. Figure 3b and c- EBSS is very effective for short periods but in panel c the authors use 12 hrs. This should be repeated with shorter times to eliminate off-target effects of long-term treatment.

In Figure 4b of the original manuscript shorter EBSS treatments were used, showing Fn14 degradation over time. Nevertheless, we decided that this line of research lies outside the main focus of this study and therefore omitted it from the revised manuscript.

The nuclear background with the LC3 antibody varies a lot- is there a reason for this?

We addressed this comment – for details please see our response to comment 5 of reviewer 1.

6. Figure 3e. Under what conditions is this experiment done? The experiment was conducted under normal growth conditions, namely cells

were grown in rich medium in absence of any inhibitor. A clarification was added to the figure legend.

7. Figure 4d is an important experiment to show the accumulation of Fn14 on either autophagic or endocytic membranes. Unfortunately the data is not convincing and insufficient quality. In particular the ATG4B panel shows uneven loading of EEA1, smearing of the lanes. It is impossible to judge where the peak and co-fractionation occurs. In addition what do the authors conclude about the LC3 panel in this condition? This must be improved and repeated at least n=3.

We thank the reviewer for this comment. The experiment was done more than 3 times. In the revised manuscript we show better quality blots. Moreover, the appearance of LC3 in the ATG4BDN panel was indeed puzzling for us too. However, the appearance of LC3II in the light fraction was inconsistent (it was not detected in at least 3 different fractionations). In the results shown in the revised manuscript we show an example in which LC3II is missing in the ATG4BDN treated cells.

8. Page 8, the authors must consider that wortmannin will also inhibit the endocytic pathway. Is Fn14 internalized normally in the presence of wortmannin?

We thank the reviewer for this comment and fully agree with the notion that wortmannin inhibits endocytosis. As a matter of fact, data presented in Figure 5a indeed indicate that wortmannin treatment inhibited Fn14 internalization. In the model of the revised manuscript we indicate that prior to the GATE-16-mediated autophagy, Fn14 is internalized to endosomes, which may either be targeted to the canonical MVB pathway or to the autophagic system. Consistent with this notion is the fact that when both systems are simultaneously inhibited Fn14 is accumulated in large EEA1-positive vesicles (Figure 5d and the revised model, Figure 7).

9. In Figure 6 the authors should show EEA1 distribution with siRNA GABARAP to show the lack of staining and to control again for internalization defects. And visa versa with siRNA GATE16 should be labelled with ERGIC antibodies.

We now added these controls in Supplementary Figure 7c,d of the revised manuscript. Our results show no Fn14 localization to this endocytic marker

upon GABARAP knockdown and conversely no ERGIC localization upon GATE-16 knockdown.

Minor points

1. Abstract last sentence “autophagosome”

Done.

2. In Fig. 4c the legend says WIPI2. Can the authors clarify this? Thank you for pointing out this mistake, we have corrected it in the text to WIPI1.

Reviewer #3 (Remarks to the Author):

The TNF receptor Fn14 is a short-lived protein. The author found that autophagy contributes to the degradation of Fn14 through a selective autophagy process dependent on P62. Interestingly, the authors found that the ATG8 family proteins play differential roles in regulating the function and turnover of Fn14. Overall, the discovery is important and data are supportive. Specific points below:

Major points:

1. An important structure described in this work is the accumulation of Fn14 into big punctate compartments in the absence of all ATG8s or autophagy inhibition. It is important to further characterize this structure to clarify it is a preautophagosomal structure. For example, a thin section EM would help, and a proteinase K treatment to determine if Fn is sequestered into the autophagosome. Also it is important to clarify that whether the structure formed under different KDs are essentially the same or they are different. In agreement with the reviewer’s concern, we invested a great deal of effort aiming to characterize these structures by EM. Unfortunately, this turned out nonproductive, most likely due to the inability of different batches of the anti-Fn14 antibody to facilitate immune-EM. Our efforts included different fixation conditions as well as few attempts to use the CLEM system, all proved unsuccessful. It is also possible that the dynamic nature of these structures

(they dissolve in the absence of p62 yet proliferate in the absence of TSG101) prevents their detection in a thin EM section.

As suggested by the reviewer we performed a proteinase K protection assay to further determine the nature of the structures accumulated in the absence of Atg8s. Data presented in Figure 4d of the revised manuscript are consistent with the notion that Fn14 accumulates in endosomes that are only partially surrounded by the autophagic membrane and are therefore not protected from proteinase K.

2. Although the work suggest that the requirement of ATG8 lipidation in the regulation of Fn14, it is helpful to further clarify if the lipidation is required by rescue the siRNA treated cells with the lipidation deficient ATG8s. We thank the reviewer for raising this important comment. We overexpressed the non-lipidated form of both GATE-16 (G116A) and GABARAP (G116A) in the matching siRNA transfected cells. Data presented in Supplementary Figures 6f, 7b of the revised manuscript indicate that lipidation of these proteins is essential for their activity.

3. The author will consider modify the model in the last Figure because 1. GATE16 is not responsible for the level of Fn14. Therefore it is not consistent in Fig.7 showing that the GATE16 pathway also directs to the lysosome. 2. The overall data is not enough to support two independent pathways controlling Fn14 trafficking because KD of all ATG8s leads to the similar localization of Fn14 with GATE16 alone. This indicates that ERGIC and endocytosis may function in a same pathway but have a relationship of up and down stream. We thank the reviewer for this comment. The scheme presented in Figure 7 of the revised manuscript is substantially modified according to our new data. The reviewer's comments were all address in our new model, as follows:

Autophagy regulates the cellular level of Fn14; GABARAP-mediated autophagy accounts for the cellular levels of Fn14, whereas in its absence the receptor accumulates in the ERGIC complex. Importantly, this pool of the receptor does not signal for NF-kB. GATE-16-mediated autophagy – acting in parallel to the canonical MVB pathway – is responsible for degradation of Fn14 localized in endocytic membranes, whereas in its absence Fn14-positive

endosomes accumulate at the vicinity of autophagic membranes while maintaining NF- κ B signaling. Of note, depletion of all Atg8s by either ATG4B^{DN} mutant, or by knockdown of all Atg8s results in a mixed phenotype whereby Fn14 accumulates in the large endocytic puncta as well as in the ERGIC complex.

In the discussion section of the revised manuscript we also raise an hypothesis for the differential roles of different Atg8s discovered in this study.

4. To further substantiate the data, the author will consider also perform a double labeling of Fn14 with other ATG8s with available antibodies. Unfortunately, there are no immunofluorescence-competent antibodies available for all Atg8 family members. We therefore stably expressed GFP-GABARAP and GFP-GATE-16 proteins in Hela cells (Figure 3e and Supplementary Figure 3) and immunostained endogenous LC3 (Figure 3a-c), showing partial colocalization with Fn14.

5. The author will consider further discuss the possible scenario about why the ATG8 homologues are different in regulating the Fn14.

We believe that different cargos are recruited into autophagosomes by a complex machinery involving ubiquitin (and potentially other post-translational modifications), autophagic receptors (such as p62 in this system) and different Atg8s. Identification of the exact ubiquitin chains (or any other modification) taking place on the receptor or its associated protein TRAF2 will help clarifies this issue in the future. A paragraph describing these hypothetical scenarios was added to the Discussion section of the revised manuscript.

Minor points:

1. Does the perinuclear puncta of Fn14 in ATG8s all KD colocalize with the ERGIC or GRASP65?

No colocalization of Fn14 and GRASP65 following knockdown of all Atg8 was detected. This control was added to Supplementary Figure 7c of our revised manuscript.

2. In Fig.3e and 6d, why the LC3 vesicles are not hollow under the STORM microscope?

The resolution in STORM is the range of 20-30 nm. In the images specified by the reviewer we show small basal autophagic bodies forming under normal growth conditions. Since the IF staining of Fn14 and the tagged Atg8 is accomplished by both a primary and secondary antibody, our resolution is decreased while looking at small vesicles, limiting our ability to detect a hollow gap.

3. There is a typo in Fig.4C. WIPI1 should be WIPI2 according to the text.

Done

4. Does KD of GABARAP or LC3C affect NF-Kb signal?

In Supplementary Figure 6g we show that the knockdown of GABARAP does not affect NF-kB signaling.

5. Fig5a, the author should use other VPS34 specific inhibitors as well as KD ATG14 to confirm.

We now added a control using an additional VPS34 inhibitor (VPS34-IN2) to Supplementary Figure 5a of our revised manuscript, which shows similar results to that of wortmannin. Unfortunately, our results using ATG14 knockdown were inconclusive and therefore excluded from the revised manuscript.

Reviewers' comments:

Reviewer #1 (Remarks to the Author):

The authors addressed all my concerns. The manuscript is excellent.

Reviewer #2 (Remarks to the Author):

The authors have overall improved the data presented to support the model in Figure 7. In addition, the authors have addressed many of my comments satisfactorily except for one. This concerns Figure 4b, the subcellular fractionation experiment. The replacement blots are not substantially improved and are inconsistent with the original figure. The original concerns about EEA1, uneven loading and the lack of conclusive data to support the localization of any of the markers remains and in fact is even more concerning now. Furthermore, if the experiments have been done more than 3 times there should be some consistent sedimentation value for the compartments. The peak of Fn14 is not at the same density as the original figure 4d (original between 28 and 31% sucrose and revised between 32 and 41%), and there are similar inconsistencies with the other markers EEA1 and p62. The peak of EEA1 is in fact shifted compared to Fn14, p62 and TTAF2. One potential issue is that EEA1 is not the best marker to use for early endosomes as most of it is cytosolic. While it is true that Fn14, p62 and TRAF2 do co-sediment at 39-41% sucrose in Fig 4b, in the original Figure 4d the co-sedimenting pool was at 30-32% sucrose. In addition, EEA1 is still not conclusive. Therefore the data in Figure 4b does not add any support for the model that Fn14 accumulates in both endocytic and autophagic structures. The remaining data is suggestive but not conclusive.

Reviewer #3 (Remarks to the Author):

The authors have appropriately addressed my concerns.

Reviewer #2 (Remarks to the Author):

The authors have overall improved the data presented to support the model in Figure 7. In addition, the authors have addressed many of my comments satisfactorily except for one. This concerns Figure 4b, the subcellular fractionation experiment. The replacement blots are not substantially improved and are inconsistent with the original figure. The original concerns about EEA1, uneven loading and the lack of conclusive data to support the localization of any of the markers remains and in fact is even more concerning now. Furthermore, if the experiments have been done more than 3 times there should be some consistent sedimentation value for the compartments. The peak of Fn14 is not at the same density as the original figure 4d (original between 28 and 31% sucrose and revised between 32 and 41%), and there are similar inconsistencies with the other markers EEA1 and p62. The peak of EEA1 is in fact shifted compared to Fn14, p62 and TTAf2. One potential issue is that EEA1 is not the best marker to use for early endosomes as most of it is cytosolic.

While it is true that Fn14, p62 and TRAF2 do co-sediment at 39-41% sucrose in Fig 4b, in the original Figure 4d the co-sedimenting pool was at 30-32% sucrose. In addition, EEA1 is still not conclusive. Therefore the data in Figure 4b does not add any support for the model that Fn14 accumulates in both endocytic and autophagic structures. The remaining data is suggestive but not conclusive.

We thank the reviewer for this comment. To address it, we re-established the cells expressing Atg4B^{DN} cells and performed a series of new experiments analyzing the effect of the Atg4B^{DN} mutant on Fn14 subcellular fractionation pattern. In figure 4b of our revised manuscript we show representative data of our experiments. As requested by the reviewer we also added to the analysis the pattern of Rab7 and Rab5, two endosomal markers. Please note that the subcellular fractionation of Fn14 in these experiments fit perfectly with that shown in our original submission.

REVIEWERS' COMMENTS:

Reviewer #2 (Remarks to the Author):

The authors have fully addressed my concerns.